# Women’s Involvement in Steady Exercise (WISE): Study Protocol for a Randomized Controlled Trial

**DOI:** 10.3390/healthcare11091279

**Published:** 2023-04-29

**Authors:** Irene Ferrando-Terradez, Lirios Dueñas, Ivana Parčina, Nemanja Ćopić, Svetlana Petronijević, Gianfranco Beltrami, Fabio Pezzoni, Constanza San Martín-Valenzuela, Maarten Gijssel, Stefano Moliterni, Panagiotis Papageorgiou, Yelko Rodríguez-Carrasco

**Affiliations:** 1Department of Physiotherapy, University of Valencia, 46010 Valencia, Spain; 2Physiotherapy in Motion, Multi-Specialty Research Group (PTinMOTION), Department of Physiotherapy, University of Valencia, 46010 Valencia, Spain; 3Faculty of Sport, University “Union—Nikola Tesla”, 11000 Belgrade, Serbia; 4University of Parma, 43121 Parma, Italy; 5Federazione Italiana Triathlon, 00135 Roma, Italy; 6Unit of Personal Autonomy, Dependency and Mental Disorder Assessment, INCLIVA Biomedical Research Institute, 46010 Valencia, Spain; 7Centro de Investigación Biomédica en Red de Salud Mental (CIBERSAM), Instituto de Salud Carlos III, 28029 Madrid, Spain; 8Kinetic Analysis, Jheronimus Academy of Data Science, 5211 DA ‘s-Hertogenbosh, The Netherlands; 9European Culture and Sport Organization (ECOS), 00183 Roma, Italy; 10European Platform for Sport Innovation (EPSI), 1000 Brussels, Belgium; 11Department of Food Science and Toxicology, Faculty of Pharmacy, University of Valencia, Av. Vicent Andrés Estellés s/n, 46100 Burjassot, Spain

**Keywords:** health promotion, exercise, patient adherence, sedentary behavior, mobile applications

## Abstract

Background: Physical inactivity is a serious public health problem for people of all ages and is currently the fourth highest global risk factor for mortality. The transition period from adolescence to adulthood coincides with a marked reduction in participation in physical activity, with more than 50% (and up to 80%) of young adults stopping physical activity. This decrease in physical activity is more evident in women than in men. Despite efforts, existing programs face challenges in effectively initiating and maintaining physical activity among individuals, particularly women, for extended durations. To address these limitations, the Women’s Involvement in Steady Exercise (WISE) randomized controlled trial (RCT) seeks to assess the efficacy of a digital high-intensity training intervention complemented by nutritional plans and other health-related advice. Methods: The study will be a three-center, randomized (1:1), controlled, parallel-group trial with a six-month intervention period. A total of 300 participants will be recruited at three study sites in Spain, Serbia and Italy. The participants will be randomized to one of the two groups and will follow a six-month program. The primary outcome of the study is the daily step count. Self-reported physical activity, the adherence to the exercise program, body composition, physical activity enjoyment, quality of sleep and physical capacities will also be evaluated.

## 1. Introduction

Physical inactivity is a serious public health problem for people of all ages and is currently the fourth highest global risk factor for mortality [1]. A lack of physical activity is related to pathologies such as type 2 diabetes, coronary heart disease and certain types of cancer, among others [2]. In addition, in the case of children and adolescents, several studies have shown that those who perform greater physical activity have better physical and mental health and better psychosocial well-being than those who lead a sedentary lifestyle [3].

It is estimated that 23% of the adult population and 81% of adolescents (between 11–17 years) do not meet the physical activity recommendations of the World Health Organization (WHO) [4]. These indications suggest that children and adolescents between 5 and 17 should perform at least 60 min a day of moderate-to-vigorous-intensity exercise, mostly aerobic, as well as incorporating strength/impact activities at least three days a week. On the other hand, the recommendations for adults between the ages of 18 and 64 are that they should perform at least 150–300 min of moderate-intensity aerobic physical activity, or at least 75–150 min of vigorous-intensity aerobic physical activity, combined with muscle-strengthening activities at moderate intensity or greater on two or more days a week [5].

Statistics from the European Union (EU) member states reveal that 60% of individuals aged 15 and above rarely or never participate in exercise or sports, and over 50% seldom or never engage in other forms of physical activity such as cycling, walking, gardening or household chores [6]. The transition period from adolescence to adulthood coincides with a marked reduction in participation in physical activity, with more than 50% (and up to 80%) of young adults stopping physical activity [7]. This decrease in physical activity is more evident in women than in men. According to the data provided in 2020 by the Survey of Sports Habits in Spain, in annual terms, only 53.9% of women practiced sport in 2020 compared to 65.5% of men. The difference between men and women is most evident in younger age groups: 15% of men aged 15–24 never exercise or play sports compared to 33% of women in the same age range [8].

Evidence reports that the best way to combat the consequences of a sedentary lifestyle is to exercise. However, one of the great problems of sports practice is the lack of adherence [9], that is, the lack of continuity to perform some type of physical activity. Barriers to physical activity and sports participation for women include a lack of time, lack of interest or low motivational level, societal norms and expectations related to gender roles and appearance, lack of social support, fear of injury, limited access to exercise facilities or prohibitive costs of training programs, cultural norms and stereotypes and personal beliefs and attitudes, among other factors. These barriers may vary by age, ethnicity and socio-economic status [10,11,12,13,14].

In order to overcome these barriers, different strategies have been described, such as group training, supervised training and different, more “dynamic” exercise modalities such as those that combine moderate aerobic exercise with strength exercise or high-intensity functional training [15]. One of these interventions is High Intensity Interval Training (hereinafter HIIT). This training modality is acquiring great acceptance as one of its main advantages is that it is a form of exercise of short duration (the sessions usually last between 20 and 30 min). However, despite its short execution time, its effects are maintained in the long term (up to 48 h after having completed the training [16]). Furthermore, it has been seen in some studies that participants prefer this type of exercise to a traditional exercise program because they enjoy it more and find it more motivating [17]. A review of HIIT programs has shown that a range of HIIT protocols, including those that use cycling, running and bodyweight exercises, can enhance cardiorespiratory fitness and other health markers in both healthy individuals and those with chronic diseases such as diabetes and hypertension [18]. Other studies have suggested that HIIT programs involving Tabata-style intervals (i.e., exercise periods that involve performing an exercise at maximum intensity for 20 s, followed by 10 s of rest, and repeating this cycle for a total of four minutes) can be effective for improving fitness in inexperienced populations [19,20]. However, it is worth noting that a gradual progression and individualized programming are crucial to prevent injury and ensure long-term success with a HIIT program.

The evidence also reports that new technologies could be a great tool to combat the aforementioned adhesion problems. Some studies have shown that new technologies, such as phone apps or smartwatches, can be a useful support to improve exercise adherence [21,22]. The Xiaomi Mi Band smartwatches have shown good accuracy in the measurement of steps [23,24], although more studies are needed in this regard as there is a lack of literature on the subject. Leveraging behavior change theories and techniques (BCTs) is a critical factor in enhancing the effectiveness of e-health interventions as they enable the targeting of key components for behavior change [25]. The Consolidated Standards of Reporting Trials (CONSORT) statement [26] and the World Health Organization (WHO) [27] have stressed the importance of incorporating a theory-based approach in the creation of digital interventions, according to their recommendations.

On the other hand, nutrition is well-recognized as a central component of a healthy lifestyle. Maintaining a healthy diet is a challenge for adolescents and young adults [28]. Dietary behaviors also tend to worsen during early adulthood, when young individuals transition into independent living [29]. Dietary behavior change programs are needed that appeal to large numbers and diverse types of people who could benefit from the educational/behavior change procedures. Innovative approaches to dietary change are needed to engage participants in enjoyable experiences to reach the largest number of participants.

For these reasons, in order to engage young women in physical activity habits, it is necessary to create a new perspective of promoting exercise. New ways of encouraging physical activity should be included. These should include a combination of new technologies with shorter and more varied exercise sessions that facilitate women’s enjoyment of physical activity and appear as a new way of improving their adherence to exercise within a healthier lifestyle.

### Study Aims

The primary goal of the Women’s Involvement in Steady Exercise (WISE) randomized controlled trial (RCT) is to evaluate the adherence to a six-month HIIT program, accompanied by nutritional plans and other health-related advice, delivered via a mobile application, among sedentary young women aged 15 to 24 years. This trial aims to determine the effectiveness of the exercise intervention by assessing the changes in the participants’ daily step count over the six-month period. The secondary objectives include analyzing the medium- and long-term effects of the program on various variables, including physical activity, anthropometric measurements, body composition, physical capacities, well-being and psychological mediators.

## 2. Materials and Methods

### 2.1. Study Design

The WISE experimental RCT consists of three components embedded into a smartphone app: (1) a remote HIIT program with video sessions; (2) an interface that includes health information; (3) an activity monitoring tool. The study will be a three-center, randomized (1:1), controlled, parallel-group trial with a six-month intervention period. Young women will be recruited from the communities at three study sites in Spain (University of Valencia, Valencia), Serbia (University Nikola Tesla, Belgrade) and Italy (SPORTLAB, Parma). This study protocol will be reported in accordance with the Standard Protocol Items: Recommendations for Interventional Trials (SPIRIT) guidelines [30], and will follow the CONSORT (Consolidated Standards for Reporting Trials) guidelines for the transparent reporting of parallel group randomized trials [31]. To gain a comprehensive understanding of the study design, refer to Figure 1.

### 2.2. Participants

#### 2.2.1. Eligibility Criteria

Participants eligible for the trial must comply with all of the following:

1.Young women aged between 15 and 24 years.2.Sedentary young women who do not comply with the WHO recommendations for physical exercise and with a low international physical activity questionnaire (IPAQ) score, which means that they do not perform at least one of these:
Three or more days of vigorous activity for at least 20 min a day.Five or more days of moderate-intensity activity.Walk at least 30 min a day every day.Five or more days of combining activities of moderate or vigorous intensity or walking reached a minimum of 600 MET (min/week).

The presence of one of the following criteria will lead to the exclusion of the participant:1.Young women with diabetes.2.Young women with possible heart problems or another type of contraindication that does not allow physical exercise (for this item the physical activity readiness questionnaire (PAR-Q) survey is used).3.Young women who are not willing to wear the smartwatch during the six-month intervention.4.Young women who have contracted severe COVID-19 prior to the intervention. Severe COVID-19 is defined as dyspnea, a respiratory rate of 30 or more breaths per minute, a blood oxygen saturation of 93% or less, a ratio of the partial pressure of arterial oxygen to the fraction of inspired oxygen (Pao2:Fio2) of less than 300 mm Hg, or infiltrates in more than 50% of the lung field [32].

#### 2.2.2. Recruitment

A total of 300 participants (150 per group) will be recruited between the three countries. Each country will recruit 100 young women for the study. Dissemination of the project will be conducted via email and posters in close contact with universities, schools and local authorities. Through the information of the posters and emails, interested participants will contact the investigators and will be informed about the study in order to decide if they finally want to participate. Those willing to participate will be summoned for a personal interview to check if they meet the inclusion criteria. Participants who meet the inclusion criteria will be invited to participate in the study and will sign a written consent form before being included in the study. At the start of the study, a wearable device (Xiaomi Mi Band 5) will be distributed to all participants.

#### 2.2.3. Sample Size and Power Analysis

Sample size estimations were derived from the primary outcome measure of daily step count, measured using the Xiaomi Mi Band 5. A priori sample size calculation was conducted based on a previous meta-analysis [33], which reported an effect size of d = 0.51 (95% CI 0.12 to 0.91, I² = 90%), compared with control groups, for physical activity interventions that included wearables and smartphone apps. Our sample size estimation was based on an intermediate effect size of d = 0.50.

To detect a difference equivalent to an effect size of 0.5 on the primary outcome with 90% power and a two-sided type I error of 0.05, a sample size of 240 is required. Specifically, assuming that the statistical unit is an individual day and considering an intraclass correlation coefficient of 0.5 to account for interindividual and intraindividual variability, a total of 2002 individual days per group (equivalent to 22 participants per group) are necessary. We suggest including a total of 300 participants (150 per group) in order to protect against a potential dropout of 20%, inherent to such a trial.

#### 2.2.4. Randomization, Allocation and Blinding

After the first experimental visit (T0), the participants will be randomized using a computer-generated randomization list into one of the two conditions with a 1:1 allocation. The allocation will be transmitted using numbered (0 = control/1 = Intervention), opaque and sealed envelopes, which will be given to the young women and then registered by the data collectors. The assessors will be blinded to the treatment allocation, although double blinding is not applicable in interventions of this nature as it is impossible to conceal the allocation from the participants. The participants will be instructed not to reveal their allocation to the assessors to ensure unbiased assessment.

### 2.3. Intervention

#### 2.3.1. Procedure

All participants will undergo a total of four visits, including the initial inclusion visit and three experimental visits (T0, T1 and T2), as detailed in Table 1. During the selection visit, qualified personnel will make sure that the young women fulfil the inclusion criteria and that they (or their parent/legal representative in case the girl was a minor) sign the informed consent form after explaining the project itself and after responding to all of their questions. One week after the initial visit, the T0 experimental visit will be conducted to measure all of the baseline assessments before the start of the intervention. At the halfway mark of the program (i.e., three months in), a mid-term measurement will be carried out, denoted as T1. The T2 experimental visit will be scheduled at the end of the program (i.e., after six months of intervention). The three experimental sessions will be identical.

#### 2.3.2. Intervention Group

In order to promote behavior change, we implemented 12 BCTs within the intervention program. Table 2 displays how the BCTs have been implemented within the WISE intervention group.

The WISE intervention is composed of four main features:Exercise videos: participants in the intervention group will receive two HIIT video sessions a week via the WISE app (Figure 2A). These sessions will be carried out at their own home without the need of sport equipment. The exercise program will be developed by two experts in physical activity and sports sciences. Each session will consist of four exercises, and, from session 10, with two different intensities so that the participants can choose one according to their physical condition. The exercises will be different every week. The second session will always have a greater intensity than the first, and the intensity will be increased from week to week. Each session will last between 20 and 30 min and will follow the following structure: 5–7 min of warm-up; 10–15 min of HIIT training; 5–7 min of stretching. Table 3 shows the progression for the WISE exercise protocol. The complete WISE exercise protocol with links to the videos of the sessions can be seen in Appendix A.

2.Education in healthy habits: once a week, the participants will receive reading material, with an approximate duration of 2 to 5 min, on general advice on nutrition, sleep (Figure 2B) and well-being (Figure 2C), written by experts in the field. Dietary guidelines will also be created by a team of professionals specialized in nutrition to stimulate the participants and give them basic indications on proper eating behavior. These reading materials will be delivered to the participants through the WISE application and will appear by means of a push notification. The complete list of messages for the WISE participants can be seen in Appendix A.3.Activity tracker: participants will be able to check their activity at any time of the day through the Xiao Mi Band 5 smartwatch (Figure 2D). The data will also be displayed in the app with weekly charts so participants can see their progression. The participants will also be able to track their weight and their sleep over time (Figure 2E) through the application. Moreover, their heart rate will be recorded. This tool aims to give feedback on behavior and promote self-monitoring of physical activity.4.Motivational activities: in order to motivate the participants to perform the videos, social media, email, WhatsApp groups and/or other channels, such as Viber (depending on the availability/popularity in each country), will be created for the participants to communicate with each other. In addition, social meetings will be organized through “Open Days” held in each country and transmitted through streaming so all the participants can attend the event. During the “Open Days”, talks on nutrition and physical activity will be held, as well as in-person meetings to perform the exercises. Moreover, the participants can expose their doubts and concerns, which can be solved in a personal manner during these events.

#### 2.3.3. Control Condition

The participants in the control group will benefit from general physical activity recommendations (at the start of the intervention: general physical activity guidelines) and will also have access to the activity monitoring tool supplied by the smartwatch. The content of both groups is summarized in Table 4.

#### 2.3.4. Adverse Events

Throughout the experiment, adverse events will be monitored for each procedure. Any adverse events or reactions that are thought to be causally associated with the intervention will be recorded and managed.

### 2.4. Outcome Measures

#### 2.4.1. Primary Outcomes

The primary outcome of this study will be the change in daily steps from baseline to three months and six months. Daily steps will be assessed using the Xiaomi Mi Band 5 smartwatch, a reliable wearable activity tracker that has been validated in previous research [35,36,37]. Days with missing data, defined as days with fewer than 1000 steps, based on previous literature [38,39], will be excluded from the analysis [40,41] to ensure accurate and comprehensive data representation.

#### 2.4.2. Secondary Outcomes

The secondary outcomes include changes in: (1) physical activity; (2) anthropometric measurements and body composition; (3) physical capacities; (4) well-being. In addition, psychological mediators will also be examined.

The following criteria will be used in order to measure the WISE program adherence [42]:Retention (completion): the participants following the WISE exercise videos and showing up to the follow-up measurements.Attendance: percentage of videos completed of the total of 50 videosDuration: adherence to a minimum of 20 min of exercise two times a week.Intensity: intensity levels of the sessions will be assessed at the end of each session using the modified Borg Scale, which measures perceived exertion [43].

These items will be measured through a weekly online exercise diary, where the participants will be asked if they completed both video sessions. They will also be asked about their perceived exertion after each session. This will allow us to assess whether they are following the 20 min of exercise two times a week rule, completing the video sessions, so we can see the number of videos completed at the end of the program.

Table 1 presents the assessment schedule, following the Standard Protocol Items: Recommendations for Interventional Trials (SPIRIT) schedule template. Table 5 provides an overview of all the outcome measures.

## 3. Statistical Analyses Plan

All the statistical analysis will be performed using the SPSS version 24 Software (SPSS Inc., Chicago, IL, USA). If adherence issues occur, all analyses will be evaluated by intention-to-treat principles, with a level of significance of 0.05. Continuous outcomes will be presented using mean and standard deviation if they follow a normal distribution. On the other hand, count outcomes, such as WISE sessions completed, will be reported through the median and percentage of achievement. At the basal assessment, an independent-samples Student’s *t*-test will be performed to rule out differences between groups of age and height of participants.

In order to answer the first aim of this study, the changes in the adherence variables, such as daily steps, the Borg scale and the IPA questionnaire, will be analyzed by 1-within-subject factor multivariate analysis of variance (MANOVA). The Bonferroni adjustment will be used for post-hoc comparisons between the T0, T1 and T2 times assessments. On the other hand, the secondary outcomes of the study will be tested by 2-factor mixed MANOVA to analyze the effects of a within-subject factor (time assessment from T0 to T4), the between-subject factor (group) and their interaction. As mentioned above, Bonferroni adjustment will be used for times and groups comparisons. In addition, the same statistical test will be carried out considering the country of origin of the data as a covariate, and the findings will be reported according to their statistical significance. For all the statistical analyses, differences will be declared statistically significant if the *p*-value is less than 0.05. The exact *p*-value and 95% confidence interval will be reported.

## 4. Ethics and Dissemination

The WISE RCT adheres to the Helsinki declaration principles. The research protocol has been reviewed and approved by the Human Research Ethics Committee of the University of Valencia (protocol number 1944476). The study is registered in the ClinicalTrials.gov (NCT05467280). Written informed consent will be obtained from each participant. All of the investigators, the ethics committees and the trial registry will be informed of any modifications that must be made to the study protocol. The results will be disseminated through international conference presentations and in relevant scientific journals.

## 5. Conclusions

This study aims to investigate the effectiveness of an online HIIT program with nutritional plans and other health-related advice in improving exercise adherence and various health outcomes among sedentary young women. A significant contribution of this study is the development of guidelines to promote exercise engagement in this population.

We expect that the WISE program will improve the participants’ knowledge of physical activity, proper nutrition and healthy habits. The program’s physical activity component will involve monitoring the participants’ heart rate, number of steps and body composition, including fat percentage, visceral fat, water and muscle mass. The diet and healthy habits component will offer basic guidelines and advice on healthy eating, quality sleep and adequate water intake, all delivered via a mobile application.

However, the study design has some limitations, including the program’s duration of six consecutive months, which includes vacations and local holidays, as well as the challenge of ensuring that the participants adhere to the program, such as wearing the smartwatches. Depending on the results obtained, the WISE program may consider adjusting the sessions to be more individualized. Additionally, as noted by DiPietro et al. [63], it is important to acknowledge that the WHO recommendations used as a filter in this study are general guidelines for the population as a whole and may not be specifically tailored to certain subpopulations, such as sedentary young women. Finally, increasing the sample size will enable the development of a specification equation (via regression analysis) to determine the general health status of the women in each age group.

## Figures and Tables

**Figure 1 healthcare-11-01279-f001:**
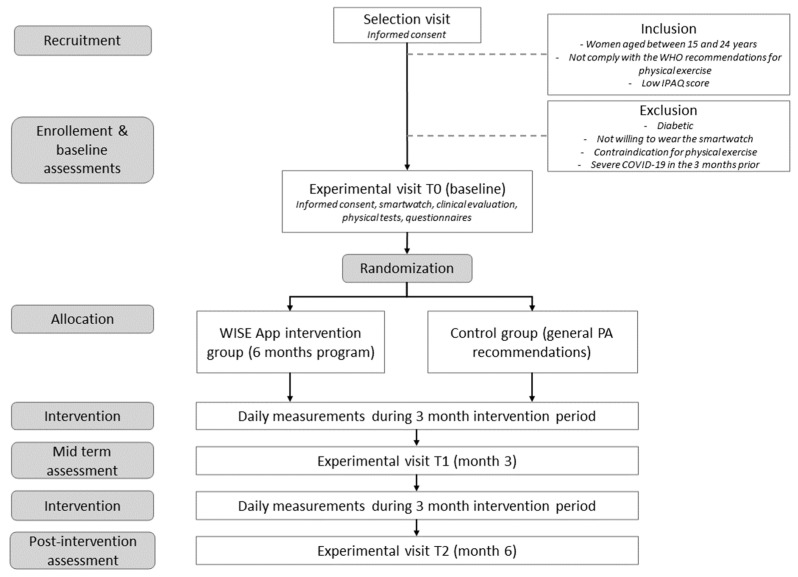
Study flow chart. WHO, World Health Organization; IPAQ, international physical activity questionnaire; PA, physical activity.

**Figure 2 healthcare-11-01279-f002:**
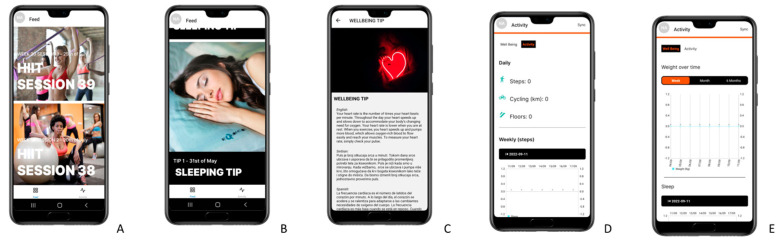
Screenshots examples of the WISE application. (**A**) HIIT video sessions. (**B**) Sleeping tips. (**C**) Well-being tips. (**D**) The activity monitoring tool. (**E**) The weight and sleep monitoring tool.

**Table 1 healthcare-11-01279-t001:** Timeline for enrolment, interventions and assessments.

	Study Period
	Selection Visit	T0	Intervention
TIMEPOINT	M-_1_	0	M_1_	M_2_	M_3_/T1	M_4_	M_5_	M_6_/T2
ENROLMENT:								
Eligibility screen	X							
Informed consent	X							
Randomization		X						
INTERVENTIONS:								
Intervention group		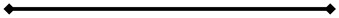
Control group		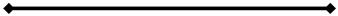
ASSESSMENTS:								
Step count		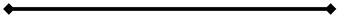
Adherence		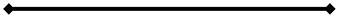
IPAQ		X			X			X
Height		X						
Weight		X			X			X
Body composition		X			X			X
Plank test		X			X			X
HLPCQ		X			X			X
PSQI		X			X			X
PACES		X			X			X
Period pain		X			X			X
CPM		X			X			X
6MWT		X			X			X

IPAQ, International Physical Activity Questionnaire; HLPCQ, Healthy Lifestyle and Personal Control Questionnaire; PSQI, Pittsburgh Sleep Quality Index; PACES, Physical Activity Enjoyment Scale; CPM, Conditioned Pain Modulation; 6MWT, 6 Minutes Walking Test. Continue line: duration over time.

**Table 2 healthcare-11-01279-t002:** BCT implementation in the intervention group following Michie et al.’s taxonomy [34].

BCT	Implementation in the WISE RCT
Goal setting behavior (1.1)	Set monthly step goal
Feedback on behavior (2.2)	Feedback on daily steps via the activity monitoring tool included in the application with weekly graphs
Self-monitoring of behavior (2.3)	Weekly diary used to document whether participants performed the exercise video and if they were engaged in any additional exercise during the study period
Feedback on outcome(s) of behavior (2.7)	Bioimpedance sheet with all the information of the changes in their body composition after 3 and 6 months of exercise
Social support (practical) (3.2)	Social days created so the participants can exercise together in real life
Social support (emotional) (3.3)	Promote social interaction through social media groups
Instruction on how to perform a behavior (4.1) Behavioral practice/rehearsal (8.1) Demonstration of the behavior (6.1)	Exercise videos
Information about health consequences (5.1)	Tips and information about the benefits of exercise and a healthy lifestyle given through the application
Prompts/cues (7.1)	Push notification through the application
Graded tasks (8.7)	Exercise program with increased difficulty

BCTs, behaviour change techniques; WISE, Women’s Involvement in Steady Exercise; RCT, randomized controlled trial.

**Table 3 healthcare-11-01279-t003:** WISE exercise protocol progression.

Month	Session	Work Time	Recovery Time	Level
1	1	20”	20”	Initiation and no impact
	2			Medium-advanced with impact
2	1	20”	15”	Initiation and no impact
	2			Medium-advanced with impact
3	1	20”	10”	Initiation and no impact
	2			Medium-advanced with impact
From week 12th, we introduce 2 different intensity levels in each session (one without impact or less demanding) so the participants can choice between the 2 levels in order to better adapt their effort				
4	1	30”	20”	Medium
	2			Advanced
5 and 6	1	40”	15”	Medium
	2			Advanced

**Table 4 healthcare-11-01279-t004:** Summary of the groups’ content.

Intervention Group (Videos)	Control Group
50 HIIT video exercise sessions	Physical activity recommendations (at the start of the intervention: general physical activity guidelines) *
Communication group	--
Activity monitoring tool (mobile app + Xiaomi Mi Band 5)	Activity monitoring tool (Xiaomi Mi Band 5)
Education in healthy habits	--

HIIT, High Intensity Interval Training. * At the end of the program, the participants will have access to the HIIT video sessions.

**Table 5 healthcare-11-01279-t005:** Outcomes measures of the WISE RCT.

Outcome	Assessment Method
Primary outcome	
Daily step count over 6 months	The Xiaomi Mi Band 5 smartwatch will be used to assess the daily step count. Days with missing data and/or days with fewer than 1000 steps will be excluded from the analysis.
Secondary outcomes	
Physical activity	
Self-reported physical activity and sedentary behaviors	Self-reported behaviors will be gathered using the International Physical Activity Questionnaire (IPAQ). The IPAQ assesses walking and activities of a moderate and vigorous intensity that are performed continuously for at least 10 min in all domains of everyday life (i.e., leisure, occupational, household and transport) in the last 7 days. The IPAQ demonstrates acceptable levels of test-retest reliability and fair to moderate associations with accelerometer measures [40,41].
Anthropometric data and body composition	
BMI, Body mass and height	Using a calibrated digital scale, body mass is measured to the nearest 0.1 kg; height is measured to the nearest 0.1 cm using a wall-mounted stadiometer. The BMI is automatically calculated by the body composition analyzer as the ratio of body mass (kg) to height squared (m²).
Body composition	Assessment of body composition is conducted using bioelectrical impedance analysis with the multi frequency segmented body composition analyzer InBody 230 (InBody, Cerritos, CA, USA) [44].
Physical capacities	
Muscle strength	The plank test protocol requires participants to maintain a static prone position with only forearms and toes touching the ground. Proper form requires feet together with toes curled under the feet, elbows forearm distance apart, and hands clasped together against the floor mat. Participants maintain eye contact with their hands, a neutral spine, and a straight line from head to ankles. The test begins when the participant demonstrates the correct position. Participants are allowed to deviate from the correct position once and can continue the test if they immediately resume the correct starting position. The test is terminated on the second deviation from the correct position or if the participant does not return to the correct position after the first warning [45]. The plank test protocol is a reliable test [46] and has been used with children [45] and young adults [46].
Endurance	The 6 Minute Walk Test measures aerobic capacity and endurance through sub-maximal exercise. The outcome by which to compare changes in performance capacity is the distance traveled during a period of 6 min. Reference equations have been developed for healthy young adults [47], and have proved Excellent test-retest reliability, interrater reliability and intrarater reliability for different populations [48,49,50,51,52].
Well-being	
Quality of life	The Healthy Lifestyle and Personal Control Questionnaire (HLPCQ) will be used to examine several dimensions of daily living. The HLPCQ has been described as a good tool for assessing the efficacy of future health-promoting interventions to improve individuals’ lifestyle and well-being. This questionnaire is a 26-item tool in which the respondent is asked to indicate the frequency of adopting 26 positively stated lifestyle habits using a Likert-type scale (1 = Never or rarely, 2 = Sometimes, 3 = Often and 4 = Always) [53].
Quality of sleep	Quality of sleep will be assessed with The Pittsburgh Sleep Quality Index (PSQI), a self-administered questionnaire used to evaluate sleep quality during the past month. The validity of the PSQI has been confirmed by several studies in different patient populations and languages [54,55,56,57]. The PSQI consists of seven clinically derived components that assess sleep difficulty, and the sum of these seven component scores yields a global score of subjective sleep quality [57]. The PSQI demonstrates moderate convergent validity compared to measures of insomnia and fatigue and good divergent validity with measures of daytime sleepiness and circadian phase preference in young adults [58].
Period pain	Pain intensity will be measured using the visual analogue scale (VAS). A 100-mm line bounded by “no pain” on the left (0) and “worst pain possible” (100) on the right will be used to indicate the average pain during the period. The minimum clinically detectable difference (MCID) of this scale has been set at 30 mm [59]. It is also a scale that has been validated [60].
Psychological mediators	
Perceived enjoyment	The Physical Activity Enjoyment Scale (PACES) is used to evaluate perceived enjoyment of physical activity during the intervention. The questionnaire comprises 16 items, where participants rate their feelings about the physical activity they have been engaging in using a 7-point Likert scale, ranging between 1 (not at all) and 7 (very much) [61]. PACES shows a very high internal consistency (Cronbach’s alpha = 0.908) and the test-retest reliability indicates a good temporary agreement (Spearman rho = 0.815, *p* < 0.001) in adolescents with overweight and obesity [62].
Program adherence	
Retention, Attendance, Duration and Intensity	These four variables will be used to measure program adherence, as described before.

WISE, Women’s Involvement in Steady Exercise; RCT, randomized controlled trial; BMI, body mass index; PPT, pressure pain threshold.

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
