# Peer review of "Women’s Involvement in Steady Exercise (WISE): Study Protocol for a Randomized Controlled Trial"

_healthcare, 2023, doi:10.3390/healthcare11091279_

Round 1

Reviewer 1 Report (Previous Reviewer 1)

Thank you for including the details regarding the intensity of HIIT sessions, and the pop-up messages regarding breaks in training regimen or substitution with low intensity exercise. I especially like the Christmas messages that do not advise the young women not to indulge, but instead advise how to mitigate negative effects of seasonal indulgences. I really hope that this research proves fruitful and results in more active young women.

Author Response

Reviewer 2 Report (New Reviewer)

Thank you very much for letting me review your article. All the comments that I am going to make are intended to improve the content of the article and the scientific quality.

Introduction.

I miss some reference like DiPierto et al. 2020. It would be interesting to note whether the general recommendations for the population are sufficient in the second paragraph

line 86.It would be advisable to define the HIIT type since this type of training was created for high performance athletes and the general population is not usually able to reach the required intensities. Even less if the exercises are strength.

Intervention

Line 207. The legend HLPCQ does not appear in the legend of the table

It should be noted that the selected 6MWT test is not intended to assess healthy young people. I understand the simplicity of this test, but it is possible that excessive changes in physical condition may not be observed, despite good adherence to the program. On the other hand, the plank test gives information about the resistance of the trunk muscles. Perhaps it would be good as a measure of strength to select another simple test such as an SJ or CMJ jump.

The program and the app are very well planned.

Author Response

Reviewer 3 Report (New Reviewer)

Congratulations for the work presented.

I would start by asking you to clarify whether the type of study will be, for example, a quasi-experimental one. In the methodology itself, it would be important to safeguard the existence or not of other exercise programs in progress, availability of free spaces and/or practice accompanied by exercise. it is difficult to guarantee a blind study design given that there will always be access to the wristband and the collection of data associated with its use. Would it be easier to have two physically different places, one doing control and the other doing cases?

Greetings, good job.

Author Response

Reviewer 4 Report (New Reviewer)

It is recommended to expand the barriers that hinder the practice of physical activity and sports in the female population, carrying out a broader explanation and with a greater contribution of updated references.

Regarding the objectives, make a better definition of them to clarify the purpose of the study, as well as its contribution. Therefore, it is recommended to expand the main objective, as well as to reformulate the secondary objectives so that they describe in a more detailed way the variables that are going to be analyzed.

It is recommended to expand the conclusions section and include the expected results of the program, as well as its strengths and weaknesses.

Author Response

This manuscript is a resubmission of an earlier submission. The following is a list of the peer review reports and author responses from that submission.

Round 1

Reviewer 1 Report

This should be a very interesting study. However, as a road cyclist who engages in HIIT training, I am skeptical that previously sedentary young women will adhere to this regimen for six months (if HIIT >= 90% aerobic capacity). HIIT training is grueling if done correctly. I really hope that young women will discover the euphoric after-effect of HIIT, but I question the effectiveness of 6 months of uninterrupted HIIT. A week of rest or low-intensity exercise interspersed throughout those six months is suggested.

While I'm not an exercise physiologist, I am a competitive road cyclist who works with a professional trainer. I engage in HIIT regularly, but as part of a training block of a few months followed by a week or two of recovery (for physical, psychological, and emotional renewal). IMHO this recovery period is critical to avoid burnout and injury, and preserve enthusiasm.
